# SDM-RL: Steady-State Divergence Maximization for Robust Reinforcement Learning

## Abstract

While reinforcement learning algorithms have achieved human-level performance in complex scenarios, they often falter when subjected to perturbations in test environments. Previous attempts to mitigate this issue have explored the training of multiple policies with varied behaviors, yet these efforts are compromised due to suboptimal choices in diversity measures. Such measures often lead to training instability or fail to capture the intended diversity among policies. In this research, we offer a unified perspective that ties together previous work through the common framework of maximizing divergence between steady-state probability distributions induced by different behavioral policies. Most importantly, we introduce an innovative diversity measure, simply used as an intrinsic reward, that addresses the limitations of prior work. Our theoretical advancements are complemented by experimental evidence across a diverse set of benchmarks.

## 1 Introduction

Reinforcement learning (RL) (Sutton & Barto, 2018) in the recent past has shown to perform better at complex tasks (Lillicrap et al., 2015) and to even exceed human level performance (Mnih et al., 2013b; 2015; Silver et al., 2016) leveraging the use of deep neural networks as function approximators. However, these RL approaches have exhibited susceptibility to even minor perturbations within test environments. One potential solution for this problem is to train the RL algorithms to be robust under those perturbations in the training phase itself (Zhang et al., 2021a; Sun et al., 2023). However, the lack of knowledge about the permissible perturbations in the test environment renders such training approaches restrictive and effective only when prior information about the test environment is available. This leads us to the question on *can we formulate a general form of robustness without any knowledge about the test environment?*

One potential solution to this question is to induce a set of policies that are different from each other. Recent works have attempted to accomplish this by characterizing the diversity within the policy set as an intrinsic reward function, optimizing it concurrently with the environmental reward so that they can induce policies that are both close to the optimal policy and different from each other. These diversity measures in existing literature can be broadly categorized into two groups: information-based diversity measures (Kumar et al., 2020; Eysenbach et al., 2018; Osa et al., 2022) that maximized the mutual information between the policies and the states encountered while executing those policies and successor feature (Barreto et al., 2018a) based diversity measures (Zahavy et al., 2022; 2023) that estimates the distance between the steady-state probability distributions (SSD) induced by different polices using successor features. As we show in this work all of these diversity measures can be seen as a relaxation or an approximation for the different approaches to measuring the divergence between the SSDs induced by polices in this set.

However, these measures have their limitations. They directly depend on the current policy, which can lead to training instability or as in the case of successor feature-based measures, struggle to converge to the true distance between steady-state distributions (SSDs) induced by different policies when environmental knowledge is lacking, resulting in weak estimation of the distance between the SSDs.

In this work, we propose what could be considered an ideal diversity measure, but demonstrate that it is not immune to limitations identified in previous work. To address this, we introduce a novel alternative measure, which we theoretically show to approximate the ideal measure more closely

than existing ones. Crucially, our alternative mitigates the stability issues seen in prior research, often attributable to the direct relationship between the diversity measure and the current policy.

Our contribution can be summarized as follows

▷ We propose an ideal diversity measure and a surrogate diversity measure for the ideal measure which addresses the limitations of both the ideal measure and the measures from the past works present.

▷ In the discrete state space setting, we theoretically show that our proposed measure is closer to the ideal diversity than the past works.

▷ On an experimentation level in both continuous and discrete control settings we show that our method performs better compared to the past works.

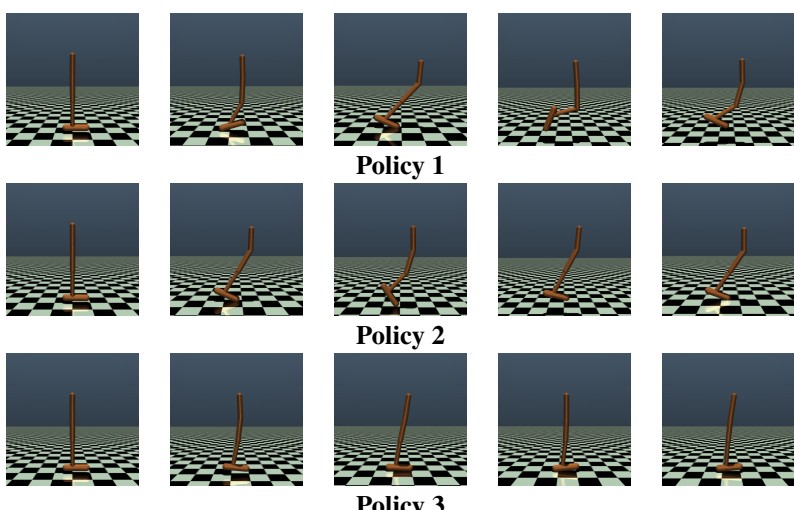

**Policy 1**

**Policy 2**

**Policy 3**

Figure 1: Examples of different near optimal polices that were learned on the hopper environment using our proposed measure. Here the learned polices 1, 2 and 3 correspond to *hopping forward while lowering the body*, *hopping only using the tip of the front foot* and *balancing in place* respectively.

## 2 RELATED WORKS

**Robustness in Reinforcement Learning** The pursuit of robustness within reinforcement learning has undergone extensive investigation, with applications spanning safety and generalization, among others. A prevalent approach to addressing robustness in reinforcement learning involves framing it as a two-player zero-sum game, wherein the adversary's action space encompasses the uncertainties present in the test environment, such as the transition function and reward function (Bagnell et al., 2001; Nilim & Ghaoui, 2004; Iyengar, 2005). An alternative avenue to induce robustness is by perturbing either the state observations (Zhang et al., 2021b; Pattanaik et al., 2018) or the agent's actions (Sun et al., 2023; Klima et al., 2019), or by accounting for uncertainties in environmental parameters like mass, friction, and gravity (Pinto et al., 2017; 2016). However, it's important to note that these methods require knowledge about the permissible range of perturbations that can be applied to the environment in test time.

**Diversity in Reinforcement Learning** In the absence of the knowledge about the uncertainty in the test environment a set of works tries to generate different close to optimal behaviours in the training in order to have robustness in test time. These line of works are closely related to our work. (Kumar et al., 2020; Eysenbach et al., 2018; Osa et al., 2022; Zahavy et al., 2022; 2023) uses a diversity measure as an auxiliary intrinsic reward to the main environmental reward. Variation of these concepts have also been used into multi agent reinforcement learning (Wang et al., 2020; Liu et al., 2017) to induce different roles in different agents.

**Steady State Probability Ratio Estimation** Estimating the steady state probability ratio between an expert policy and a learners policy has been a complex task due to their dependency on both the policy and the policy's interaction with the environment. There have been methods that estimate this ratio (Gelada & Bellemare, 2019; Hallak & Mannor, 2017) using the steady-state property of stationary distributions of Markov processes. But these methods come with the limitation of needing a explicit knowledge of the policy making it impossible to learn in the face of data from a mixture of polices or non markovian behaviour polices. To this end recent works (Nachum et al., 2019; Kostrikov et al., 2019) have proposed a policy agnostic estimation of steady state probability ratios. Our diversity estimation objective in Equation 3 falls on to a similar type of ratio estimation.

## 3 PRELIMINARIES

### 3.1 REINFORCEMENT LEARNING

We consider a Markov decision process (MDP), defined by the tuple $< S, A, p, r >$, where the state space $S$ and the action space $A$ which is either continuous or discrete, and the unknown state transition probability $p(s_{t+1}|s_t, a_t)$ represents the probability of the next state $s_{t+1} \in S$ given the current state $s_t \in S$ and action $a \in A$. The environment gives a bounded reward $r(s_t, a_t)$ which is bounded by $[R_{min}, R_{max}]$ on each transition. We will use $\pi(a_t|s_t)$ to denote the policy that dictates the action taken by an agent given a state. As a general notation given a probability space $X$, we will use $x \in X$ to indicate the random variable from the space and $x_i \in X$ as the realization of the random variable. We will also use $x_t$ to indicate a time-indexed random variable. We will define the total accumulated reward during an episode of length $H$ given a start state $s_0$ by $J_\pi(s_0) = \sum_{t=0}^{t=H} r_t$. The maximum possible reward an agent can get in the environment is given by $R_{\max}$. The steady-state distribution that is induced by a certain policy $\pi_i$ is defined as $d_{\pi_i}$. This can either be a steady state distribution $d_{\pi_i}(s)$ or a steady state-action probability distribution $d_{\pi_i}(s, a)$.

### 3.2 DIVERSE REINFORCEMENT LEARNING

When it comes to diverse reinforcement learning the policy is defined by a set of policies $\pi \in \Pi$ where each realization of the policy $\pi_i$ is a separate policy and the goal of the learner is to learn a set of policies such that each of those policies is close to optimal and diverse enough from the other policies. In general, the cardinality of the set of policies is set as a non-infinite number. Given a set of latent variable $Z$ and a certain $z_i \in Z$ we use $\pi_i^{-1}$ to define the collection of all policies in the set other than the policy $\pi_i$ corresponding to the latent variable $z_i$. In situations where we don't have information about all the policies in the set, we will slightly abuse this notation and use $\pi_i^{-1}$ to define the collection of all the available policies other than $\pi_i$. Once the training is done in general a random $z_i$ is drawn from a probability $p(z)$ and the corresponding policy $\pi_i$ is used as the current policy. In practice, $z_i$s are drawn from a uniform distribution.

In general, the diversity is either implicitly or explicitly defined by a measure of how separate the agent's policy is on the steady state probability distribution space. The diverse policies are obtained by explicitly maximizing the diversity measure along with the reward. Generally, these methods can be classified into two major categories as follows.

▷ Information based methods (Kumar et al., 2020; Eysenbach et al., 2018; Osa et al., 2022): Here the information between the latent variable $z$ and either the states visited by a policies or the trajectories generated by the policies is used as a diversity measure. To be exact, either $I(s; z)$ or $I(\tau; z)$ is used as the diversity measure where $\tau$ denotes the trajectory and $I(.)$ refers to the mutual information. As we see later this type of setting can also eventually be derived as an implicate measure of the distance between the steady state probability distribution induced by these policies.

▷ Successor Feature (SF) Based (Zahavy et al., 2022; 2023): Here the direct objective is to characterize distance between the SSDs induced by the polices in the set of polices as the diversity. But instead of directly approximating the SSDs the expected return of successor features (Barreto et al., 2018b) $\psi = \mathbb{E}_{s,a \sim d_\pi(s,a)}[\phi(s, a)]$ are used as an estimate for the actual steady state probability distribution. Here $\phi_{i+1}$ denotes the successor feature and $\psi$ denotes the expected successor feature. As seen later this surrogate function can be a weak estimate in most scenarios.

We discuss about these methods in detail in Section 5.

In most instances, the diversity measure serves as an auxiliary reward in conjunction with the environmental reward. When the diversity measure or function is dependent on the current policy $\pi_i$, it has the potential to escalate to significantly higher values, since the learning algorithm strives to discover policies that yield greater diversity values and it can now choose a policy such that the diversity function yields higher values in general for all states as it now has the ability to change the auxiliary reward function itself. In case of the information based methods this dependency is convex in the steady state probability induced by the policy $\pi_i$ as shown in A.1. Consequently, in such scenarios, the pursuit of diversity may lead to policies with extremely low rewards as the diversity value surpasses a certain threshold. To mitigate this, an indicator constraints as below is normally used confine the influence of diversity beyond a certain threshold. The general objective for a diverse policy optimization of the past works (both the successor feature and information based methods) can be summarized as follows.

$$J_{\pi_i} = \mathbb{E}_{(s,a)\sim d_{\pi_i}}[r_t(s,a) + \alpha_{\mathsf{div}}.\mathbb{1}_{(\kappa.R_{\max}<R_{\pi_i})}\mathsf{Diversity}_{\pi_i,\pi_i^{-1}}(s,a)] \tag{1}$$

Here $\mathsf{Diversity}_{\pi_{z_i},\pi_{z_i}^{-1}}(s,a)$ refers to the diversity between a policy $\pi_i$ and the rest of the policies in the set. Here $R_{\max}$ is the maximum reward on the training environment. $R_{\pi_i}$ refers to the return of the current policy where $0 < \kappa < 1$ and $\mathbb{1}$ is an indicator random variable. However, employing such an indicator random variable can potentially cause all policies to converge towards the optimal one, unless the parameter $\kappa$ is chosen judiciously. Intuitively, selecting a higher value for $\kappa$ implies that most state transition tuples $(s,a,r,s')$ will lack the diversity term in reward, thereby driving the algorithm to optimize all policies towards a singular, optimal policy, devoid of diversity. Conversely, opting for a lower $\kappa$ value may result in an overall performance decline. Furthermore, in offline setting due to the presence of such an indicator random variable the replay buffer will contains samples following two different objectives (when the indicator random variable is zero vs non zero) this would make the learning for the agent difficult as it can lead to insufficient support for a specific diverse policy within the replay buffer. Moreover, tuning of this $\kappa$ also require some level of intuition about the ideal reward in the environment. Thus the removal of the diversity function's dependence on the policy $\pi_i$ can help us alleviate these optimization related problems.

## 4 PROPOSED DIVERSITY MEASURE

We propose our diversity measure to evaluate the overall diversity in the set of polices $\Pi$ as follows

$$\mathsf{Diversity}_{\pi_i^{-1}}^{\mathsf{ours}}(.) = E_{z_i\sim p(z)}[E_{(s)\sim d_{\pi_i}}(\log\frac{\mathsf{Uniform}(s)}{d_{\pi_i^{-1}}(s)})] \tag{2}$$

The essence of this diversity measure lies in its reliance on the steady state probability distribution of other policies $\pi_i^{-1}$ as an indicator to identify states that are frequently visited. Consequently, it can be used as a diversity measure to encourage the optimization of different policies while having optimality in mind. In fact, within a constant factor, this measure is equivalent to maximizing the entropy of the steady-state probability $d_{\pi_{i-1}}$. But the direct computation of the entropy involves estimating the $d_{\pi_{i-1}}$ probability itself. But the formulation of Equation 2 as KL divergence allows us to use the Donsker-Varadhan representation (Donsker & Varadhan, 1983) of KL divergence to estimate the KL divergence using samples by minimizing the following objective.

$$-D_{\mathrm{KL}}\left(d_{\pi_i}\|d_{\pi_{i-1}}\right) = \min_{\nu_i:\mathcal{S}\to\mathbb{R}}\log\mathbb{E}_{(s)\sim d_{\pi_{i-1}}}\left[e^{\nu_i(s)}\right] - \mathbb{E}_{(s)\sim\mathsf{Uniform}(s)}[\nu_i(s)] \tag{3}$$

If we use a function approximator $\nu_i : S \to \mathbb{R}$ the optimal solution to the above objective will give us the estimate of the term needed to evaluate the KL divergece as follows.

$$\nu_i^*(s) = \log\frac{\mathsf{Uniform}(s)}{d_{\pi_{i-1}}(s)} + C \tag{4}$$

A key observation about this diversity measure $\text{Diversity}^{\text{ours}}_{\pi^{-1}_i}(.)$ is its independence from the current policy $\pi_i$, distinguishing it from previous approaches. This unique characteristic renders the diversity function a constant with regard to the current policy, effectively mitigating the issue of diversity escalation. Consequently, we can express the policy optimization objective as follows, eliminating the need for the indicator random variable $\mathbb{1}(\alpha.R_{\max} < R\pi_i)$. The pseudo code for the algorithm can be found on Appendix A.5.

$$J_{\pi_i} = E_{(s,a) \sim d_{\pi_i}}[r_t(s,a) + \alpha * \text{Diversity}^{\text{ours}}_{\pi_{i-1}}(s,a)] \tag{5}$$

We can use any standard reinforcement learning algorithm to optimize this objective. Now, we only need to adjust $0 < \alpha_{\text{div}} < 1$ instead of having to fine-tune both $\kappa$ and $\alpha$ as was required in prior works. This effectively addresses the limitations that were encountered in previous optimization approaches. Furthermore, the objective in Equation 3 is also suited to measuring the diversity between non Markov behaviours and Markov polices. Thus this method can also be generalized for setting where the diversity in polices is induced with an expert behaviour in mind where the trained agents maybe working with an expert in the test settings. Even though this is beyond the scope of this paper the ability to do this also gives our work more validity in terms of generalizability.

# 5 JUSTIFICATION OF OUR PROPOSED DIVERSITY MEASURE

## 5.1 IDEAL DIVERSITY MEASURE

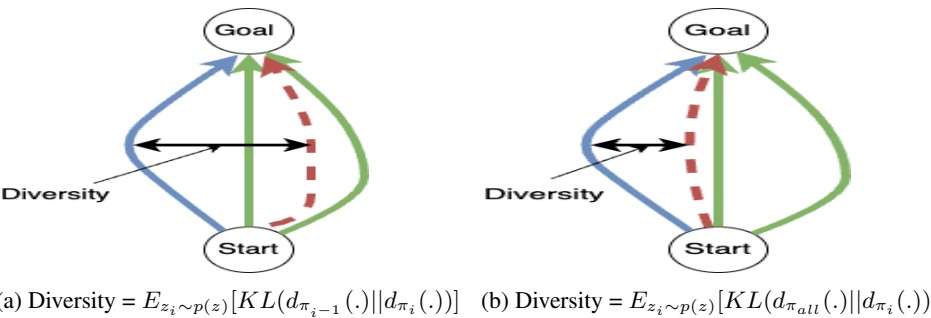

(a) Diversity = $E_{z_i \sim p(z)}[KL(d_{\pi_{i-1}}(.)||d_{\pi_i}(.))]$     (b) Diversity = $E_{z_i \sim p(z)}[KL(d_{\pi_{all}}(.)||d_{\pi_i}(.))]$

Figure 2: The figure gives a geometrical example on why the ideal diversity measure is better than the information diversity measure. In this example of reaching the goal from the start state the green paths denote the a set of polices denoted by $\pi_i^{-1}$ while the blue path denotes the policy $\pi_i$. The red line denote the average polices. In case of (a) is it the average of only the $\pi_i^{-1}$ polices and in case of (b) it is the average of all three polices $\pi_{all}$. If we are to compute the diversity between the current policy and the average policy given by the red line we can see that the ideal diversity measure (a) calculate an accurate diversity measure than the non ideal measure (b)

In this section, we propose a diversity measure as follows, which we argue as an ideal diversity measure against the existing works in the following sections.

$$Diversity^{ideal}(.) = E_{z_i \sim p(z)}[KL(d_{\pi_{i-1}}(.)||d_{\pi_i}(.))] \tag{6}$$

$$= E_{z_i \sim p(z)}[E_{(s) \sim d_{\pi_i}}(\log \frac{d_{\pi_i}(s)}{d_{\pi_{i-1}}(s)})] \tag{7}$$

However, it is evident that this measure or function still depends on the current policy $\pi_i$ underscoring the necessity for an optimization objective akin to Equation 1. This is what motivated us to alternatively propose a diversity measure in Equation 2 independent of the current policy $\pi$. In the following sections we show that the proposed measure in Equation 2 is still theoretically better than the past works.

## 5.2 INFORMATION BASED METHOD

Information-based methods typically define diversity as the mutual information between the latent variable $z$ and either the trajectory or the state. This intuitively couples different trajectories or state visitations with distinct realizations of $z$. When the information between $z$ and the trajectory equals zero, it signifies that the choice of $z_i$ has no impact on the trajectory. Conversely, if this information is higher, it implies that different policies are dependent on the specific values of $z$. With generally used assumption that $z$s are sampled from a uniform distribution, we can equivalently express the diversity measure as $KL(d_{\pi_z}(.) || \sum_k d_{\pi_k}(.))$, up to a constant factor. Here, we slightly abuse notation to denote the mixture of steady distribution probability induced by all the policies in the set $\Pi$ as $\sum_k d_{\pi_k}(.)$.

$$Diversity^{info}(.) = KL(d_{\pi_z}(.) || \sum_k d_{\pi_k}(.)) \tag{8}$$

$$= E_{z_i \sim p(z)}[E_{(s) \sim d_{\pi_i}}(\log \frac{d_{\pi_i}(s)}{\sum_k d_{\pi_k}(s)})] \tag{9}$$

Information-based methods differ from the ideal diversity measure in that the ideal measure maximizes the divergence between the steady-state probability distribution induced by the current policy and the steady-state distribution induced by all the other policies. In contrast, the information-based method maximizes the divergence between the steady-state probability induced by the current policy and all policies, including itself. This difference is one of the reasons why we contend that the ideal measure offers a better separation between the current policy and the other policies in the set.

**Theorem 1.** *In discrete state-space settings, the difference between the ideal diversity measure and the information-based diversity measure, $\delta_{info}$, is always smaller than the deviation between the proposed diversity measure and the ideal diversity measure, $\delta_{ours}$.*

*Proof.* **Proof Sketch**: Since these measures involve divergences, we can lowerbound the information based measure's closeness to the ideal diversity measure , as follows, where $C$ is a constant and $H(.)$ denotes entropy.

$$\delta^{info} \geq E_{z_i \sim p(z)}[H(d_{\pi_i}(.), d_{\pi_{i-1}}(.)) - H(d_{\pi_i}(.))] - C \tag{10}$$

Similarly we can measure proposed measure's Equation 2 deviation from the ideal measure as follows with a constant $K$

$$\delta^{ours} = E_{z_i \sim p(z)}[-H(d_{\pi_i}(.))] + K \tag{11}$$

The differences between $\delta_{info}$ and $\delta_{ours}$ while ignoring the constants can be written as

$$\delta^{info} - \delta^{ours} = E_{z_i \sim p(z)}[H(d_{\pi_i}(.), d_{\pi_{i-1}}(.))] \tag{12}$$

In the case of discrete state space the cross entropy is always non negative. Thus $\delta^{info} \geq \delta^{ours}$ concluding the proof. ☐

Theorem 1 makes our proposed measure closer to the ideal measure than the information based measure. This on top of the independence from the current policy gives a superiority for our measure against the information based measure.

For a detailed derivation, please refer to the Appendix A.1, A.2, A.3, A.4.

## 5.3 SUCCESSOR FEATURE BASED METHOD

Successor feature based methods also work on the idea of maximising the steady state probabilities between the polices but instead of working directly on the probabilities they use the expected successor features as a surrogate measure. The idea of successor features is to decompose the rewards as a linear combination of some feature vector of choice. Here the feature $\phi$ is known as the successor feature.

$$r(s, a) = \phi(s, a)^\top \mathbf{w} \tag{13}$$

The expectation of the consecutive successor features give an state and action $s, a$ is treated as an estimate of the steady state probability distribution.

$$\psi(s, a) = \mathbb{E}_{s,a \sim d_\pi(s,a)}[\phi(s, a)] \tag{14}$$

Then using the vector representation of the steady state probability distribution we can define the diversity as some distance measure for instance the Hausdorff distance.

$$\text{Diversity}_{\pi_i, \pi_i^{-1}}(s, a) = 0.5 \sum_{i=1}^{n} \min_{j \neq i} \left\| \psi^i - \psi^j \right\|_2^2 \tag{15}$$

When successor features are selected as the one-hot vector representation in the state-action space, $\psi(s, a)$ converges towards the steady-state probability distribution, as $\mathbf{w}$ effectively becomes equivalent to $r(s, a)$. However, as we deviate from the one-hot vector representation, some information from the reward itself becomes encapsulated by the successor feature, weakening the assumption of convergence to the real steady-state probability. In practice, state observation can be chosen as the feature vector. But in high-dimensional setting, raw state observations may prove insufficient for estimating diversity, necessitating the learner to have a certain degree of environmental knowledge for designing these feature vectors. This makes this measure less generalizable compared to both the ideal measure in Equation 8 and our proposed measure in Equation 2.

## 6 EXPERIMENTS

We design our experiments to answer question which we believe as central to proving our hypothesis. *Does the proposed measure induce diverse polices without collapsing to a single policy?* To this end, we evaluate our proposed metric using a 2D grid world environment and continuous control tasks (Todorov et al., 2012). In our experiments, we optimize our the diversity-based objective in Equation 5 in conjunction with Soft Actor Critic (SAC) (Haarnoja et al., 2018) as the base algorithm for continuous control and Q-learning (Mnih et al., 2013a) as the base algorithm for discrete control scenario. We benchmark our diversity measure based method (Ours) against the SAC base algorithm in the continuous domain and Q-learning in the discrete domain, along with the Information-based method (Kumar et al., 2020) and the Successor Feature-based method (SF) (Zahavy et al., 2022). We maintain the same base algorithm across all these methods for consistency.

### 6.1 GRIDWORLD:ROBUSTNESS UNDER PERTURBATION

Here as seen on Figure 3 (a) in the training environment task of the agent is to move from the grey box to the black box in as few steps as possible. In the test environment the agent is faced with obstacles of different sizes as show in Figure 3 (b,c,d) where the optimal policy possible from the training environment will not be enough successfully navigate in the test environment thus requiring the agent to generate polices that are diverse enough while sacrificing on the optimality at some level. Here the agent receives a reward of 20 when it reaches the goal and is penalized in a dynamic manner at every step based on the distance from the goal.

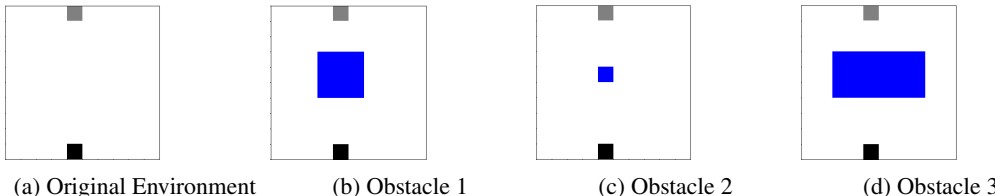

| (a) Original Environment | (b) Obstacle 1 | (c) Obstacle 2 | (d) Obstacle 3 |

Figure 3: Illustration of environment obstacles during test time. Here the grey box indicate the starting point and the black box indicates the goal. Blue boxes indicate the obstacles that were added in the new environment. Here the goal of the agent is to navigate to the black box from the grey box.

As seen in Figure 4 a setting like this where only a single optimal policy is possible the presence of the indicator random variable based filter $\mathbb{1}_{(\kappa.R_{\max}<R_{\pi_i})}$ and the diversity measure's dependence on the current policy had lead to the collapsing of polices into a single optimal policy while the current policy independent diversity measure of ours lead to a stable training thus inducing different polices which makes it robust to the unseen test environment 1. The intuition behind this observation is that when there is only one optimal policy that is possible in the environment the hyper parameter tuning $\kappa$ becomes paramount because the policies needed.

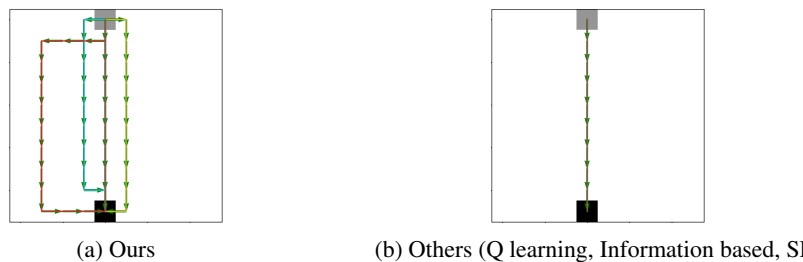

| (a) Ours | (b) Others (Q learning, Information based, SF) |

Figure 4: Demonstration of the learned diverse yet near-optimal policies. Here the lines denote the trajectories of the polices generated by each algorithm.

| Method | Original Environment | Obstacle 1 | Obstacle 2 | Obstacle 3 |
|---|---|---|---|---|
| Q Learning | **15.6** | -70.03 | -70.03 | -70.03 |
| SF | **15.6** | -70.03 | -70.03 | -70.03 |
| Information based | **15.6** | -70.03 | -70.03 | -70.03 |
| Ours | 13.47 | **-51.12** | **-6.17** | **-51.12** |

Table 1: Comparison of our proposed methods against Q Learning, Information based and the successor feature based method in the grid world environment. Here the test environment was defined with additional unseen obstacles in the training. Here both the successor feature based method and the information based method suffered from all the polices collapsing into an optimal policy resulting in them under-performing compared to our method in the test scenarios.

## 6.2 CONTINUOUS CONTROL TASKS:ROBUSTNESS UNDER PERTURBATION

In our study within the realm of continuous control, we conducted a benchmark comparison between our approach and previous methods, using the established Mujoco control tasks in the Gymnasium environment (Towers et al., 2023). Specifically, we evaluated our approach in the Hopper and Ant environments. During the training phase, our agents were trained to generate multiple policies for the standard control tasks. In the testing phase, we introduced perturbations by simulating joint malfunctions, where specific joint torques were set to fixed values. The testing tasks were defined based on these joint failures as follows.

**Ant**

▷ Test 1: Fix the torque of Ant's *right back leg* hinge joint at -1.0

▷ Test 2: Fix the torque of Ant's *front right leg* hinge joint at -1.0

▷ Test 3: Fix the torque of Ant's *back leg* hinge joint at -1.0

**Hopper**

▷ Test 1: Fix the torque of Hopper's *thigh* hinge joint at 0.5

As seen in the Table 2 and Table 3 on average our proposed method was able to experimentally perform better as opposed to the past works while sacrificing some performance in the training environment as expected.

| Method | Original Environment | Test 1 | Test 2 | Test 3 |
|---|---|---|---|---|
| SAC | **3593 ± 1115** | 55 ± 114 | 42 ±172 | -13 ± 148 |
| SF | 1553 ± 772 | 351 ± 410 | -145 ± 193 | 208 ± 220 |
| Information | 779 ± 385 | **659 ± 230** | 266 ± 56 | 302 ± 154 |
| Ours | 1335 ± 663 | 568 ± 327 | **615 ± 448** | **498 ± 512** |

Table 2: Comparison of our proposed methods against SAC, Information based and the successor feature based method. Here the test experiments were conducted by selectively disabling certain joints in the Ant. Across all tasks, our algorithm, on average, achieved a higher cumulative reward at the cost of a lower cumulative reward in the training environment.

| Method | Original Environment | Test 1 |
|---|---|---|
| SAC | **3263 ± 507** | 98 ± 47 |
| SF | 1419 ± 800 | 313 ± 31 |
| Information | 2930 ± 266 | 147 ± 71 |
| Ours | 1277 ± 567 | **435 ± 83** |

Table 3: Comparison of our proposed methods against SAC, Information based and the successor feature based method. Here the test experiment was designed by disabling thigh joint of the Hopper. As observed in previous experiments, our algorithm achieved a higher cumulative reward on the test task, albeit at the cost of a relatively lower cumulative reward in the training environment.

## 6.3 COVERAGE: CONTINUOUS CONTROL TASKS

We additionally perform a visual assessment of our proposed measure's coverage in the state space against the past works. Here we collect the states visited following all the polices generated by a particular algorithm and we perform a dimesionality reduction via PCA (F.R.S., 1901) and create a scatter plot of the points along the first two principal axes. Consistent with the findings presented in the previous section, our diversity measure effectively demonstrates superior coverage of the state space, as illustrated in Figure 5.

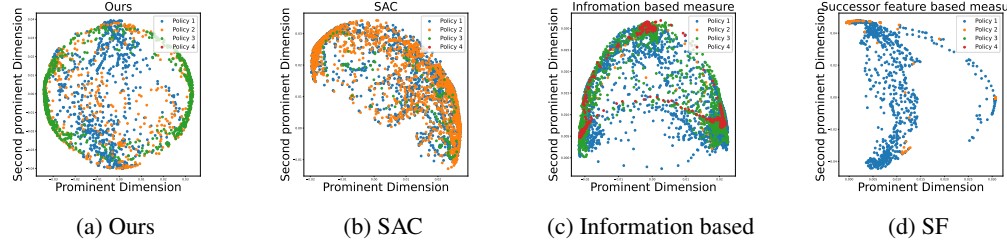

(a) Ours      (b) SAC      (c) Information based      (d) SF

Figure 5: This figure illustrates the state space coverage by the policies generated by each agent in the hopper environment. Dimensionality reduction was performed through PCA on the visited states. Here the x and y axes represent the two primary dimensions resulting from PCA. Our proposed method demonstrated better coverage compared to alternative methods.

### 6.4 Hyperparameters

In the case of SAC, the policy and value-based estimators used a network architecture with two hidden layers, each of size 256, utilizing ReLU non-linearity (Agarap, 2018). Both the alpha and the network used a learning rate of $3 \times 10^{-4}$. A discount rate $\gamma$ of 0.99 was employed, along with a soft update rate of $5 \times 10^{-2}$ to update the target value function alongside the current value function. The $\nu_i(s)$ estimator from our method, the discriminator from the information-based method, and the expected successor feature estimator all shared the same architecture with two hidden layers, each of size 256, and employed ReLU non-linearity. Their respective learning rates were $3 \times 10^{-6}$, $3 \times 10^{-4}$, and $3 \times 10^{-4}$. In both the information-based method and SF-based methods, $\kappa$ was selected from within the range of 0.2 to 0.75. For all the algorithms, $\alpha_{\mathsf{div}}$ was searched across a range of 0.1 to 10.

## 7 Conclusion

In this work, we analyse the diversity measures employed to elicit distinct behaviors in reinforcement learning agents in the existing works and identify the limitations posed by them. Additionally, we introduce an alternative diversity metric that adeptly mitigates these limitations, offering a more effective indicator of the diversity among these different behaviours. Our primary focus of this work has been on the analysis of diversity metrics. We demonstrate that our proposed metric effectively addresses these limitations, both theoretically and empirically.

One non explored avenue in this work is the proposed framework's ability to induce different behaviours with regards to the non Markovian polices. This is an interesting avenue because it can facilitate on training agents that work in complementary to an expert in tasks such as search and rescue where it will be more beneficial to train the agents to exhibit behaviours not followed by the human in order to increase the cooperative coverage of the area. This poses as an interesting direction for the future works.

Our proposed measure also has its own limitations. Specifically, it necessitates the storage of evaluation samples of policies during training to estimate the diversity measure, potentially increasing storage requirements. But this comes with an advantage that samples from some expert polices can be used without any knowledge about the expert polices while inducing diversity. Consequently, a potential avenue for future research lies in finding a balance where the diversity of current learnable policies can be efficiently evaluated without the requirement for evaluation samples, while still enabling the estimation of diversity with respect to available expert data samples.

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

# A APPENDIX

## A.1 INFORMATION BASED MEASURE DERIVATION

Given that the choice of $p_z$ for sampling $z_i$s is $p(z) \sim Unifrom(1, n_z)$ a catergorical uniform distribution where the cardiality of $|Z| = n_z$ we can write the mutual infromation based measure as follows.

$$
\begin{aligned}
I(s; z) = \quad & I(z; s) \\
= \quad & \sum_{s_j, z_i} p(s_j, z_i) \log \frac{p(s_j, z_i)}{p(z_i)p(s_j)} \\
= \quad & \sum_{s_j, z_i} p(s_j, z_i) \log \frac{p(z_j|s_i)}{p(z_i)} \\
= \quad & \sum_{s_j, z_i} p(s_j|z_i).p(z_i) \log \frac{p(z_i|s_j)}{p(z_i)} \\
= \quad & \sum_{s_j, z_i} p(s_j|z_i).n_z^{-1} \log \frac{p(z_i|s_j)}{n_z^{-1}} \\
= \quad & \sum_{z_i} n_z^{-1}.\sum_{s_j, z_i} d_{\pi_i}(s) \log \frac{\tilde{d}_{\pi_i}(s)}{\sum_k d_{\pi_k}(s).n_z^{-1}} \\
= \quad & \sum_{z_i} n_z^{-1}.KL(d_{\pi_{z_i}}(.)||d_{\pi_{z_{all}}}(.)) \\
= \quad & E_{z_i \sim p(z)}[KL(d_{\pi_{z_i}}(.)||d_{\pi_{z_{all}}}(.))]
\end{aligned}
\tag{16}
$$

Here we can write $p(s_j|z_i) = d_{\pi_i}(s_j)$ as it is the steady state probability induced by the policy corresponding to a certain $z_i \in Z$. Furthermore we replace If we replace $p(z_i|s_j)$ in 16 with the following.

$$
\begin{aligned}
p(z_i|s_j) = \quad & \frac{p(s_j|z_i).p(z_i)}{\sum_k p(s_j|k).p(k)} \\
= \quad & \frac{d_{\pi_i}.p(z_i)}{\sum_k d_{\pi_k}(s_j).p(k)} \\
= \quad & \frac{d_{\pi_i}(s_j).n_z^{-1}}{\sum_k d_{\pi_k}(s_j).n_z^{-1}} \\
= \quad & \frac{d_{\pi_i}(s_j)}{\sum_k d_{\pi_k}(s_j)}
\end{aligned}
\tag{17}
$$

Since KL divergence is a convex function we can see that the information measure for the $i$th policy is convex on $d_{\pi_i}$ induced by that policy $\pi_i$.

## A.2 INFORMATION BASED MEASURE'S DEVIATION FROM IDEAL MEASURE

$$
\begin{aligned}
\delta^{\text{info}} = \quad & E_{z_i \sim p(z)}[KL(d_{\pi_i}(.)||d_{\pi_{i-1}}(.))] - E_{z_i \sim p(z)}[KL(d_{\pi_i}(.)||d_{\pi_{i_{all}}}(.))] \\
= \quad & E_{z_i \sim p(z)}[\sum_{s_j} d_{\pi_i}(s) \log \frac{d_{\pi_i}(s)}{d_{\pi_{i-1}}(s)}] - E_{z_i \sim p(z)}[\sum_{s_j} d_{\pi_i}(s) \log \frac{d_{\pi_i}(s)}{d_{\pi_{i_{all}}}(s)})] \\
= \quad & E_{z_i \sim p(z)}[\sum_{s_j} d_{\pi_i}(s_j)(\log \frac{d_{\pi_i}(s_j)}{\sum_{k \neq i} d_{\pi_k}(s).(n-1)^{-1}} - \log \frac{d_{\pi_i}(s_j)}{\sum_k d_{\pi_k}(s_j).n^{-1}})] \\
= \quad & E_{z_i \sim p(z)}[\sum_{s_j} d_{\pi_i}(s_j) \log(\frac{\sum_k d_{\pi_k}(s).n^{-1}}{\sum_{k \neq i} d_{\pi_k}(s).(n-1)^{-1}})]
\end{aligned}
\tag{18}
$$

## A.3 PROPOSED METHOD'S DEVIATION FROM IDEAL MEASURE

$$
\begin{aligned}
\delta^{\text{ours}} = \quad & E_{z_i \sim p(z)}[KL(d_{\pi_i}(.)||d_{\pi_{i-1}}(.))] - E_{z_i \sim p(z)}[KL(\text{Uniform}(.)||d_{\pi_{i-1}}(.))] \\
= \quad & E_{z_i \sim p(z)}[\sum_{s_j} d_{\pi_i}(s) \log \frac{d_{\pi_i}(s)}{d_{\pi_{i-1}}(s)}] - E_{z_i \sim p(z)}[\sum_{s_j} d_{\pi_i}(s) \log \frac{n_s^{-1}}{d_{\pi_{i-1}}(s)})] \\
= \quad & E_{z_i \sim p(z)}[\sum_{s_j} d_{\pi_i}(s_j)(\log \frac{d_{\pi_i}(s_j)}{\sum_{k \neq i} d_{\pi_k}(s).(n-1)^{-1}} - \log \frac{n_s^{-1}}{\sum_{k \neq i} d_{\pi_k}(s).(n-1)^{-1}})] \\
= \quad & E_{z_i \sim p(z)}[\sum_{s_j} d_{\pi_i}(s_j)(\log(d_{\pi_i}(s_j)) + \log(n_s))] \\
= \quad & E_{z_i \sim p(z)}[\sum_{s_j} d_{\pi_i}(s_j) \log(d_{\pi_i}(s_j))] + K \\
= \quad & E_{z_i \sim p(z)}[-H(d_{\pi_i}(.))] + K
\end{aligned}
\tag{19}
$$

## A.4 COMPARISON OF OUR DEVIATION VS INFORMATION BASED MEASURE'S DEVIATION

Since log is a monotonic function we can lower bound the information based measure's deviation as follows

$$
\begin{aligned}
\delta^{\mathsf{info}} = \quad & E_{z_i \sim p(z)}[\textstyle\sum_{s_j}(d_{\pi_i}(s_j)\log(\frac{\sum_{k \neq i} d_{\pi_k}(s).(n-1)_z^{-1})}{\sum_k d_{\pi_k}(s).n_z^{-1}}))] \\
\leq \quad & E_{z_i \sim p(z)}[\textstyle\sum_{s_j}(d_{\pi_i}(s_j)\log(\frac{\sum_{k \neq i} d_{\pi_k}(s).(n-1)_z^{-1})}{d_{\pi_i}(s).n_z^{-1}}))[ \\
\leq \quad & E_{z_i \sim p(z)}[\textstyle\sum_{s_j}(d_{\pi_i}(s_j)(\log(\frac{\sum_{k \neq i} d_{\pi_k}(s).(n-1)^{-1}}{d_{\pi_i}(s)}))] + \log(n_z) \\
\leq \quad & E_{z_i \sim p(z)}[-KL(d_{\pi_i}(.)||d_{\pi_{i-1}}(.))] + \log(n_z)] \\
\geq \quad & E_{z_i \sim p(z)}[KL(d_{\pi_i}(.)||d_{\pi_{i-1}}(.))] - C \\
\geq \quad & E_{z_i \sim p(z)}[H(d_{\pi_i}(.), d_{\pi_{i-1}}(.)) - H(d_{\pi_i}(.))] - C
\end{aligned}
\tag{20}
$$

If we compared both $\delta^{\mathsf{info}}$ and $\delta^{\mathsf{ours}}$

$$
\begin{aligned}
\delta^{\mathsf{info}} - \delta^{\mathsf{ours}} = \quad & H(d_{\pi_i}(.), d_{\pi_{i-1}}(.)) - H(d_{\pi_{z_i}}(.)) + H(d_{\pi_i}(.)) + M \\
\delta^{\mathsf{info}} - \delta^{\mathsf{ours}} = \quad & H(d_{\pi_i}(.), d_{\pi_{i-1}}(.)) + M
\end{aligned}
\tag{21}
$$

For a discrete case the cross entropy is a non negative quantity. Thus always $\delta_{\mathsf{ours}} \leq \delta_{\mathsf{info}}$ making our measure close to the ideal measure than the information based measure.

## A.5 PSEUDO CODE

---

**Algorithm 1 SDM-RL** with SAC

---

1: **procedure** INITIALIZATION
2: Initialize the ratio network $\nu_\omega$
3: Initialize the policy $\pi_{\theta_z}$
4: Initialize Q functions $Q_{\phi_1}, Q_{\phi_2}$.
5: Initialize $Q_{\phi_{tar_1}}, Q_{\phi_{tar_2}}$.
6: Set $\phi_{tar_1} \leftarrow \phi_1$ and $\phi_{tar_2} \leftarrow \phi_2$
7: Initialize the replay buffer $D_z$s for each policy
8:
9: **procedure** TRAINING
10:     **for** $z = $ no polices **do**
11:         Train the $\nu$ network with the samples $D_{z-1}$ from previous polices $\pi_i^{-1}$ using the loss function
12:         $\min_{\nu_i : \mathcal{S} \to \mathbb{R}} \log \mathbb{E}_{(s) \sim d_{\pi_{z_i^{-1}}}}\left[e^{\nu_i(s)}\right] - \mathbb{E}_{(s) \sim \mathsf{Uniform}(s)}[\nu_i(s)]$
13:         **for** $j = $ no episodes **do**
14:             Observe a state $s$
15:             **while** $s_j$ is not terminal **do**
16:                 Get an action $a = \phi(|s)$s
17:                 Implement the action and observe a next state $s'$, termination signal $d$ and a reward $r$.
18:                 Store the sample $< s, a, s', r, d >$ in the replay buffer $D$.
19:                 **if** Update **then**
20:                     Sample $B = < s, a, s', r, d >$
21:                     Compute the target Q function as
22:                     $y(r, s', d) = r + \alpha\nu(s, a) + \gamma(1 - d)\left(\min_{i=1,2} Q_{\phi_{\mathsf{targ},i}}(s', a') - \alpha\log\pi_{\theta_z}(\tilde{a}' \mid s')\right), \quad a' \sim \pi_{\theta_z}(\cdot \mid s')$
23:                     Update the Q function using the gradient
24:                     $\nabla_{\phi_i} \frac{1}{|B|} \sum_{(s,a,r,s',d) \in B} (Q_{\phi_i}(s, a) - y(r, s', d))^2 \quad$ for $i = 1, 2$
25:                     Update the policy using the gradient
26:                     $\nabla_\theta \frac{1}{|B|} \sum_{s=} \left(\min_{i=1,2} Q_{\phi_i}(s, \tilde{a}_\theta(s)) - \alpha\log\pi_{\theta_z}(\tilde{a}_\theta(s) \mid s)\right)$
27:                     Update the target networks
28:                     $\phi_{\mathsf{targ},i} \leftarrow \rho\phi_{\mathsf{targ},i} + (1 - \rho)\phi_i \quad$ for $i = 1, 2$

---

A.6    VISUALIZATION OF THE DIVERSITY MEASURE LANDSCAPE FOR THE GRID WORLD
       PROBLEM

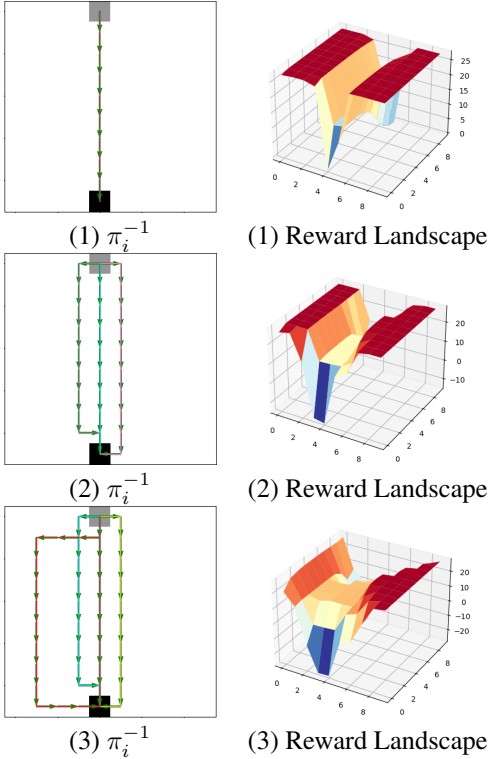

| (1) $\pi_i^{-1}$ | (1) Reward Landscape |
| (2) $\pi_i^{-1}$ | (2) Reward Landscape |
| (3) $\pi_i^{-1}$ | (3) Reward Landscape |

Figure 6: This figure illustrates learned diversity measure's landscape corresponding policy of others in different occasion in the gridworld setting. For the figures in the right the x, y axis denote the grid and the z axis denotes the value of the reward function. This figure illustrates on how visted states are penalized by a function independent of the policy $\pi_i$ under the proposed diversity measure.

## A.7    COVERAGE: CONTINUAL CONTROL TASK

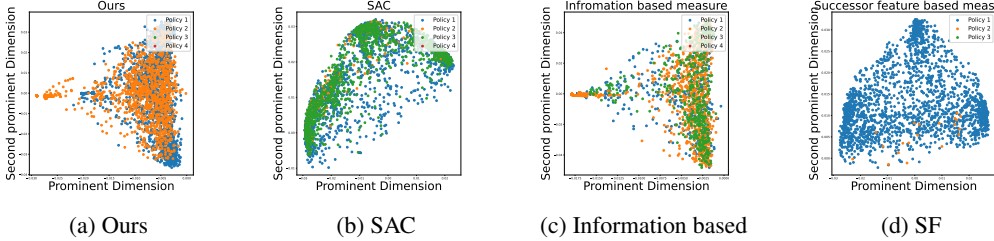

(a) Ours                (b) SAC                (c) Information based                (d) SF

Figure 7: This figure illustrates the state space coverage by the policies generated by each agent in the ant environment. Dimensionality reduction was performed through PCA on the visited states. Here the x and y axes represents the two primary dimensions resulting from PCA. Our proposed method demonstrated better coverage in the areas on the right hand side of the space a compared to the lack of coverage in case of SAC. All three of the polices showing better performance than the standard SAC on test task while having certain level of coverage in this area leads us to hypothesize that the diverse polices are generated by the visitation of these states. Our proposed measure with a higher coverage was able perform better in average the than other methods both enhancing our hypothesis and the ability of our measure to induce better diversity.

