# OpenReview forum: "SDM-RL: Steady-State Divergence Maximization for Robust Reinforcement Learning"
_ICLR.cc/2024/Conference — Submitted to ICLR 2024_

### Official Review · Reviewer_b7cc · 2023-10-30

**Soundness:** 2 fair
**Presentation:** 1 poor
**Contribution:** 2 fair
**Rating:** 3
**Confidence:** 3

**Summary:**

Classical robust MDP-based methods require knowledge about the permissible range of perturbations that can be applied to the environment in test time. To overcome it, diversity RL tries to generate different close to optimal behaviors in the training to have robustness in test time. However, previous methods are compromised due to suboptimal choices in diversity measures. This paper offers a new diversity measure, which could be used as an intrinsic reward. Finally, the authors justify the proposed diversity measure theoretically and empirically.

**Strengths:**

* The proposed diversity measure seems novel and theoretically closer to the ideal measure in the discrete state space setting

**Weaknesses:**

Quality:
* There is no standard deviation reported in experiments.
* The cumulative reward of proposed methods seems too bad in the continuous action scenario.
* Since the authors don't provide any code to reproduce results, I am a little bit doubtful about the results of SAC. Due to the property of maximum entropy, SAC was shown to solve some robust RL problems by maximizing the lower bound on a robust RL objective [Eysenbach and Levine 2022, Maximum entropy RL solves some robust RL problems].
* It is better to give empirical results under different perturbations. Additionally, more results of other discrete environment (e.g., perturbed CartPole) and MuJoCo environment experiments (e.g., HalfCheetah, Walker) are expected.

Clarity:
* Currently, related works are placed between approach and experiments. It would be better as a subsection after the introduction to give readers more background knowledge.
* In Appendix A, there is no need to repeat the LHS of the equation again and again.
* Some typos
  * Section 2.1 -> (S, A, p, r)
  * Section 2.2 no (s, a) for \psi
  * Repeated proof and proof sketch after Theorem 1

**Questions:**

Please refer to the "weakness" section for further information.

---

> ### Author Response · Authors · 2023-11-22
> **Response to  Reviewer b7cc**
>
> We thank the reviewer and the area chairs for their constructive comments and we will address the minor concerns.
>
>
>
> * I have changed the location the related work section
> * We have corrected the repetitive LHS side in the appendix.
> * We have fixed the typos
> * The cumulative reward was lower for the proposed method as expected inline with the fact that the diversity induced in the perturbed environment is relatively higher. Since the goal is to find a policy that sacrifices its performance in the training environment inorder to induce diversity so that there would be a better performance in the perturbed test environment. We find the lower reward experienced in the training environment on average to be as expected.
> * In terms of SAC it does maximizes diversity but it does so by maximizing the policy. But the other baselines and the current work does so by maximizing the steady state occupancy with regards to other policies while maximizing the entropy of the policy as well since SAC is used as the base algorithm. That is the reason for the difference in reward in the test environment when a major perturbation was done.
> * The reviewer's comments with regards to the lack of more experiments and pertubations was well taken and acknowledged.
> * With regards to the standard deviation we have avoided it because it was the standard deviation among different polices not the standard deviation of a singular policy and for a diverse set of policy it was supposed to be higher as the aim of the paper is to find polices such that certain polices perform well under certain scenarios in the expense of not being good in certain scenarios. We have added the standard deviation now.

---

### Official Review · Reviewer_YmNh · 2023-10-31

**Soundness:** 2 fair
**Presentation:** 3 good
**Contribution:** 2 fair
**Rating:** 3
**Confidence:** 4

**Summary:**

The paper present an exploration strategy based on diversity that aims at maximizing "divergence between steady-state probability distributions induced by different behavioral policies". This diversity measure is used as an intrinsic reward. The paper provides some theoretical justifications as well as some experiments.

**Strengths:**

- The paper is overall well-written
- It tackles an important topic

**Weaknesses:**

- It is slightly unclear how the method is significantly different than other similar existing techniques. In particular, the following stated hypothesis seems to be achieved with other techniques: "Does the proposed measure induce diverse polices without collapsing to a single policy".
- It is unclear how the proposed method could scale to complex environments
- Some good practice for the experiments are not followed, for instance there is no standard deviation information for the results

**Questions:**

- In Table 1, it is written that all the polices except your own collapse "into an optimal policy resulting in them under-performing compared to our method in the test scenarios.". Other techniques such as adding an entropy regularizer in the policy could already achieve better performance than all the baselines used. Are there other relevant baselines that could be considered?
- Can you add standard deviation information for the results?
- Can you provide some clarifications to elements such as "the test environment was defined with additional unseen obstacles in the training"? It seems that obstacles are added but that the state the agent perceives is the same? Otherwise you could still get the optimal policy for the training environment but due to the generalization capabilities of neural networks, it could end up with different policies on unseen tasks.

---

> ### Author Response · Authors · 2023-11-22
> **Response to Reviewer YmNh**
>
> We thank the reviewer and the area chairs for their constructive comments and we will address the minor concerns.
>
>
> *  The difference from the past works comes in the fact that the current method does have a diversity measure that is independent of the current policy as compared to previous methods which were dependent on the current policy. This can lead to the diversity function itself changing with the change in policy. When maximizing the diversity is the objective the policy can change such that the functional dependency is exploited and the diversity measure grows well beyond the reward thus resulting in diversity by suboptimal policies.
> * The point with regards to adding entropy regularizer was well taken. Within the scope of this paper we were only evaluating the effectiveness of the existing diversity measure while keeping the base RL algorithm constant. That was the reason why we didn’t consider the addition of additional entropy regularizer terms. But this is a point that was well  taken.
> * The obstacles added in case of the experiments were limited to changes in the transition probability of the environment. Thus the state space remains the same. For instance in the gridworld setting the presence of the obstacle doesn’t change the state space but only how the state transitions near the obstacle. This is an unseen transition. Similarly for the Mujoco tasks we used the transition changes in the form of perturbation on the joints.
> * The point about the scalability in complex environments is well taken. In the scope of this paper we didn’t perform experiments on complex environments with visual state inputs.
> * With regards to the standard deviation we have avoided it because it was the standard deviation among different polices not the standard deviation of a singular policy and for a diverse set of policy it was supposed to be higher as the aim of the paper is to find polices such that certain polices perform well under certain scenarios in the expense of not being good in certain scenarios. We have added the standard deviation now.

---

### Official Review · Reviewer_rZkm · 2023-11-01

**Soundness:** 2 fair
**Presentation:** 2 fair
**Contribution:** 2 fair
**Rating:** 3
**Confidence:** 4

**Summary:**

This paper proposes a method for promoting diversity via an intrinsic motivation term that maximizes the entropy of the steady-state distribution.  The paper provides a way of more easily approximating this term by rewriting the KL divergence and using function approximation.

**Strengths:**

The paper attacks an important problem of discovering more diverse policies, which is often helpful in exploring new environments. They approached the problem differently than much of the prior work, offering some diversity in the study of promoting diversity in policies.

**Weaknesses:**

The paper is often quite vague. For instance, It was hard to tell what was claimed to be "ideal" and why that would be "ideal."  Theorem 1 references the "idea diversity measure," but there is never an equation or expression named that clearly. Similarly, for the "information-based diversity" and the "proposed diversity".

Similarly, there are many equations in section 3 that are claimed to be related, but the relationships are never proven or stated formally. Where they are formal, they appear to be incorrect. For instance, equation 3 claims a direct correspondence between the KL divergence and a function approximator, which would not be true for a random neural network.  It's not clear how this function approximator factors into equation 2 since neither KL divergence nor the function approximator are listed in equation 2.

There are also many references to nitty-gritty details of other methods which are never introduced or made explicit enough to track down. For instance, the sentence before equation 5 states that it removes the need for an indicator variable, but it's not clear why that would otherwise have been needed.


A related issue to the lack of clarity is the lack of motivation. For instance, equation 2 appears quite arbitrary and is not really introduced. The justification comes later when it is claimed that it is equivalent to maximizing the entropy of the steady-state distribution but is easier to compute. If that is true, then "maximizing the entropy of the steady state distribution" should be the measure, and this should be later introduced as a way of more easily computing it.

Similarly, there is a claim in the introduction that having more policies helps robustness, but it is not clear why that would be.

**Questions:**

In what sense is maximizing entropy "ideal" when most random states of the system are useless or dangerous? For instance, most random arrangements of matter are useless, and most large random changes make society worse.

---

> ### Author Response · Authors · 2023-11-22
> **Response to Reviewer rZkm**
>
> We thank the reviewer and the area chairs for their constructive comments and we will address the minor concerns.
>
>
> * We added a geometrical justification towards why the ideal diversity measure as defined in Equation 6 proposed is better than the information based diversity measure in Figure 2. In terms of the information based diversity measure we have condensed all the past works that use information based methods into a single formulation based on steady state distribution maximization. We have added an equation additionally in section 4.2. Additionally, we have also specified the ideal diversity measure in equation
> * The function approximator that optimizes equation 2 as in equation 3 would be the estimate of the KL divergence by the Donsker Varadhan equation. We have added more text in section 3 making a statement about that.
> * The indicator random variable adds inconsistencies in terms of the samples collected. For instance when the indicator random variable becomes zero the samples will correspond to the optimal objective. Only when indicator random variable remains non zeros would the samples correspond to the diverse but close to optimal objective. But given that the diversity measure itself functionally depends on policy unless the $\kappa$ parameter is not properly tuned the diversity measure can grow beyond the range of reward at which point the policy will generate diverse by suboptimal behavior which will eventually result in the indicator random variable becoming zero and the objective effectively turning into a traditional RL objective. Thus it is hard to train diverse and close to optimal behavior in the presence of such an indicator random variable. We have added the explanation in the paper. Also if there is an indicator random variable of such nature then the replay buffer will be filled with policies following different objectives (when it is zero vs when it is not zero) which is not ideal for training as we practically saw this would lead to them learning only the optimal behavior in the absence of the enough support for a diverse close to optimal behavior.
> * Generating more policies that are different from each other and close to optimal is a paradigm that was followed by the past works (both the information based and steady state based) to induce robustness when there is a lack of any information about the test environment. While it is true that this type of paradigm doesn’t guarantee robustness in all types of test environments, it is an intuitive way to make diverse policies in the absence of any knowledge about the perturbations that can happen in the test environment.
> * The point about why entropy maximization would not necessarily be useful is well taken. Random maximization of the entropy will indeed lead to non ideal states. But in our case we are finding a balance between the reward maximization and the randomness maximization which in some sense constrains the diversity that is induced to be useful in terms of reward. As long as the safety constraints are included in the reward the entropy maximization that is done this way would avoid dangerous states and induce diverse behaviors.

---

> > ### Comment · Reviewer_rZkm · 2023-11-22
> > **Response to Reviewer Response**
> >
> > Thank you for the detailed response and changes to the manuscript.
> >
> > > We added a geometrical justification towards why the ideal diversity measure as defined in Equation 6 proposed is better than the information based diversity measure in Figure 2. In terms of the information based diversity measure we have condensed all the past works that use information based methods into a single formulation based on steady state distribution maximization. We have added an equation additionally in section 4.2. Additionally, we have also specified the ideal diversity measure in equation
> >
> > I'm still unclear as to why this is called the "ideal".  I also do not see section 4.2 in the manuscript, so perhaps I am not seeing your changes correctly. Unfortunately, it is difficult to evaluate this without seeing the referenced equations.
> >
> > >The function approximator that optimizes equation 2 as in equation 3 would be the estimate of the KL divergence by the Donsker Varadhan equation. We have added more text in section 3 making a statement about that.
> >
> > The concern is more that this equation as written and taken literally, is false, since the sides of the equation would not be equal if the function approximator is not optimal. This equality clearly must be replaced with some other symbol but I am unsure what was trying to be communicated with equation 3, so I cannot recommend a symbol.
> >
> > > The indicator random variable adds inconsistencies in terms of the samples collected.
> >
> > What method are you referring to that needs such an indicator variable?  And where in that method?  If the ability to remove this is critical to the impact of your approach it is important the reader is slowly introduced to these details
> >
> > > Generating more policies that are different from each other and close to optimal is a paradigm that was followed by the past works (both the information based and steady state based) to induce robustness when there is a lack of any information about the test environment. While it is true that this type of paradigm doesn’t guarantee robustness in all types of test environments, it is an intuitive way to make diverse policies in the absence of any knowledge about the perturbations that can happen in the test environment.
> >
> > Is the approach to try out several of these policies at test time and then pick the best? In this case it would be a sort of multi-trial robustness as opposed to a robust 1-shot performance. In any case, this should be explained to the reader rather than assumed as background as it can be clarified in a single sentence in the introduction.
> >
> > Due to remaining concerns, I will be leaving my score unchanged, but I appreciate the engagement with the points being raised.

---

### Official Review · Reviewer_Vyzf · 2023-11-10

**Soundness:** 2 fair
**Presentation:** 2 fair
**Contribution:** 2 fair
**Rating:** 5
**Confidence:** 4

**Summary:**

The paper proposes a new measure for learning an ensemble of RL policies that exhibit behavioral diversity in an MDP, while solving the task at hand or achieving near-optimal environmental returns. The proposed measure builds upon the concept of maximizing the divergence between the steady-state visitation distributions of the component policies, a notion well-explored in existing literature. In practical terms, optimizing the measure involves augmenting the environmental reward with an intrinsic reward and employing any conventional RL algorithm. The authors contrast their approach with the existing diverse RL methods based on mutual-information maximization and use of successor features. Experiments in grid-world environment and MuJoCo continuous control locomotion environments indicate that the proposed method generates diverse policies that are more robust to environmental perturbations, compared to the baseline methods.

**Strengths:**

Designing algorithms for generation of high-performing, diverse RL policies is an interesting and challenging problem, with a wide variety of practical use cases. The authors provide a good motivation for their solution by highlighting the limitations of the existing methods – namely, the dependence of existing diversity measures on the current policy, that could lead to exploitation of the diversity reward and other hyperparameter tuning challenges. The detailed exposition on prior work on information-based and successor-feature-based methods is insightful and helps to contextualize the paper’s contribution.

**Weaknesses:**

The writing and presentation of the material needs improvements in several places. There are notational inconsistencies and unclear descriptions that make certain parts hard to grasp. The experiments sections could include better interpretations and intuitions about the observed trends.

**Questions:**

1.	Please add an algorithm box in the main paper that outlines the complete algorithm. It's difficult to understand how the paper's contributions work with the deep RL algorithms otherwise.
2.	The notations $\pi_{z^{-1}_i}$ and $\pi^{-1}_i$ denote the same entity I believe, but they seem to be mixed unnecessarily at places which causes confusion while reading. For example, Equation 1 uses the former, while the next line (which is describing the equation) uses the latter. Please be consistent wherever possible.
3.	Equation 3 – the LHS seems to be incorrect. Please check the steady-state distributions inserted in the KL term.
4.	Equation 6 – check the LHS here as well. Should the distributions be put in the other order?
5.	The connection between Equation 3/4 and Equation 2 is not clear from the contents of section 3. To me, it became evident much later that you are learning a network (v) to estimate the distribution ratio: uniform(s)/d(s). Please add more details to Section 3 about this.
6.	It is claimed that the metric in Eq 2 is “independent” of the current policy. The metric still involves sampling states from the steady-state distribution under the current policy. Please properly qualify what you mean by “independence” in this context.
7.	Theorem 1 – should it be “larger” instead of “smaller” in the statement?
8.	Grid-world Experiments – please add some explanation or intuitions as to why prior methods like Kumar et al. and Zahavy et al. collapse to a single policy in such a simple MDP. This observation is quite counter-intuitive to me.
9.	All experiments – how many policies are trained simultaneously in the ensemble and what’s the effect of the ensemble-size on the algorithm? Also, it is possible to compute the overall diversity metric (Equation 2) as the training progresses and include a plot of this metric over time?

---

> ### Author Response · Authors · 2023-11-22
> **Response to Reviewer Vyzf**
>
> We thank the reviewer and the area chairs for their constructive comments and we will address the minor concerns.
>
> * We have added a pseudo code for the Algorithm in the appendix
> * We have corrected the notational errors with regards to $\pi$
> * The LHS was corrected with the correct term in equation 3
> * We have changed the ordering on the LHS in equation 6 as well
> * We have added the fact that we are estimating the KL divergence in section 3.
> * With regards to the independence of the diversity with respect to the policy we have defined the independence as follows. In practice we would like to use the estimated term $\nu$ as an auxiliary reward. The second expectation on the outside means that we are finding the expectation of that $\nu$ while we are following the current policy corresponding to $\pi_i$.  This is equivalent to how we find the reward in a traditional RL setting on expectation. The accumulated reward depends on the policy followed based on how it is accumulated via following the policy but the reward function itself (equivalent to $\nu$ in our setting) is not dependent on the policy and it remains stationary. This is the sense in which we meant that the reward is independent of the policy. We have added a clarification with regards to this.
> * When it comes to the theorem it should be smaller. Since the difference is smaller our proposed measure is always close to what we argue as the ideal diversity measure
> * In terms of the gridworld experiments we have added some intuition in the paper. The intuition is as follows. Since we have a term with the indicator random variable in the previous objective (which was needed because of the above mentioned dependance  on the policy) it can easily go to zero if the hyper parameter is not tuned properly making the training  objective similar to that of a traditional RL objective. This was very evident as in the case of the gridworld problem where there was only one optimal policy and not many.
> * The policies in the current setup were trained on after the other. That is a z is selected and a policy is trained based on the past policies. But it can be trained in parallel as well. In terms of training in parallel it accounts computing the same objective in equation 2 with respect to the data available from all other policies. The $\nu$ function in this case needs to be trained in a delayed manner similar to the hard or soft update that is done on Soft Actor Critic value functions before using them for policy optimization.

---

### Meta-Review · Area_Chair_E9JX · 2023-12-07

**Metareview:**

The paper proposes an intrinsic reward scheme to encourage learning diverse behavioral policies in RL that can robustly generalize to perturbed test environments. All the reviewers agree that the paper is below the bar for publication, because the experimental evaluation is lacking (e.g. no assessment of standard deviation/variability, unclear if appropriate baselines have been run, missing details about algorithm hyper-parameters like ensemble size). Moreover in the theoretical exposition, reviewers identified several mistakes and statements that are not logically evident, and hence more like claims but presented as proofs.

**Justification For Why Not Higher Score:**

Improving the exposition and clarity, and running a more rigorous evaluation will substantially strengthen the paper.

**Justification For Why Not Lower Score:**

N/A

---

### Decision · Program_Chairs · 2024-01-16

Reject